# GENERATIVE MODELS AS A DATA SOURCE FOR MULTIVIEW REPRESENTATION LEARNING

**Ali Jahanian, Xavier Puig, Yonglong Tian, Phillip Isola**
Massachusetts Institute of Technology
Cambridge, MA 02139, USA
{jahanian,xpuig,yonglong,phillipi}@mit.edu

## ABSTRACT

Generative models are now capable of producing highly realistic images that look nearly indistinguishable from the data on which they are trained. This raises the question: if we have good enough generative models, do we still need datasets? We investigate this question in the setting of learning general-purpose visual representations from a black-box generative model rather than directly from data. Given an off-the-shelf image generator without any access to its training data, we train representations from the samples output by this generator. We compare several representation learning methods that can be applied to this setting, using the latent space of the generator to generate multiple "views" of the same semantic content. We show that for contrastive methods, this multiview data can naturally be used to identify positive pairs (nearby in latent space) and negative pairs (far apart in latent space). We find that the resulting representations rival or even outperform those learned directly from real data, but that good performance requires care in the sampling strategy applied and the training method. Generative models can be viewed as a compressed and organized copy of a dataset, and we envision a future where more and more "model zoos" proliferate while datasets become increasingly unwieldy, missing, or private. This paper suggests several techniques for dealing with visual representation learning in such a future. Code is available on our project page https://ali-design.github.io/GenRep/.

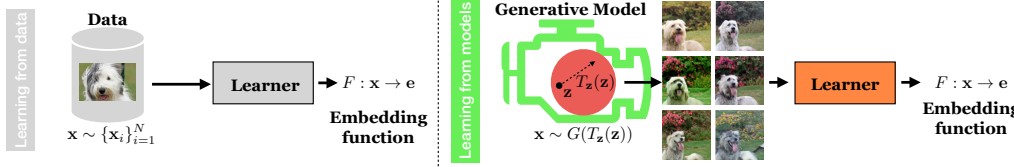

Figure 1: Visual representation learning typically consists of training an image embedding function, $F : \mathbf{x} \rightarrow \mathbf{e}$, given a dataset of real images $\{\mathbf{x}_i\}_{i=1}^N$ (left panel). In our work (right panel), we study how to learn representations given instead a black-box generative model $G$. Generative models allow us to sample continuous streams of synthetic data. By applying transformations $T_{\mathbf{z}}$ on the latent vectors $\mathbf{z}$ of the model, we can create multiple data "views" that can serve as effective training data for representation learners.

## 1 INTRODUCTION

The last few years have seen great progress in the diversity and quality of generative models. For almost every popular data domain there is now a generative model that produces realistic samples from that domain, be it images (Radford et al., 2015; Brock et al., 2019; Ramesh et al., 2021), music (Dhariwal et al., 2020), or text (Brown et al., 2020). This raises an intriguing possibility: what we used to do with *real* data, can we now do instead with *synthetic* data, sampled from a generative model?

If so, there would be immediate advantages. Models are highly compressed compared to the datasets they represent and therefore easier to share and store. Synthetic data also circumvents some of the concerns around privacy and usage rights that limit the distribution of real datasets (Tucker et al., 2020; DuMont Schütte et al., 2021), and models can be edited to censor sensitive attributes (Liao

et al., 2019), remove biases that exist in real datasets (Tan et al., 2020; Ramaswamy et al., 2020), or steer toward other task-specific needs (Jahanian et al., 2020; Goetschalckx et al., 2019; Shen et al., 2020). Perhaps because of these advantages, it is becoming increasingly common for pre-trained generative models to be shared online without their original training data being made easily accessible. This approach has been taken by individuals who may not have the resources or intellectual property rights to release the original data[1] and in the case of large-scale models such as GPT-3 (Brown et al., 2020), where the training data has been kept private but model samples are available through an API.

Our work therefore targets a problem setting that has received little prior attention: given access *only* to a trained generative model, and no access to the dataset that trained it, can we learn effective visual representations?

To this end, we provide an exploratory study of representation learning in the setting of synthetic data sampled from pre-trained generative models: we analyze which representation learning algorithms are applicable, how well they work, and how they can be modified to make use of the special structure provided by deep generative networks.

Figure. 1 lays out the framework we study: we compare learning visual embedding functions $F$ from real data $\mathbf{x} \sim \{\mathbf{x}_i\}_{i=1}^{N}$ vs. from generated data $\mathbf{x} \sim G$ controlled via latent transformations. We study generation and representation learning both with and without class labels, and test representation learners based on several objectives. We evaluate representations via transfer performance on held out datasets and tasks.

For representation learners, we focus primarily on contrastive methods that learn to associate multiple "views" of the same scene. These views may be co-occurring sensory signals such as imagery and sounds (e.g., (de Sa, 1994)), or may be different augmented or transformed versions of the same image (e.g., (Becker & Hinton, 1992; Chen et al., 2020c)). Interestingly, generative models can also create multiple views of an image: by steering in their latent spaces they can achieve camera and color transformations (Jahanian et al., 2020) and more (Härkönen et al., 2020; Wu et al., 2020). Figure 2 diagrams the currently popular setting, where views are generated as pixel-space transformations, versus the setting we focus on, where views are generated via latent-space transformations.

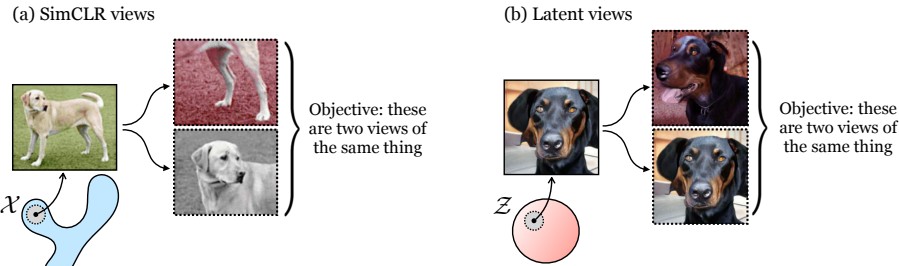

Figure 2: Different ways of creating multiple views of the same "content". (a) SimCLR (Chen et al., 2020c) creates views by transforming an input image with standard pixel-space ($\mathcal{X}$) data augmentations (example images taken from (Chen et al., 2020c)). (b) With a generative model, we can instead create views by sampling nearby points in latent space $\mathcal{Z}$, exploiting the fact that nearby points in latent space tend to generate imagery of the same semantic object. Note that these examples are illustrative, the actual transformations that achieve the best results are shown in Fig. 7.

We study the properties of these latent views in contrastive learning and ask under what conditions they can lead to enhanced learners. We then compare these learners with non-contrastive encoders, also trained on samples from generative models. Further, we study the long-standing promise of generate models in being capable of generating more samples than the number of datapoints that trained them. Knowing current models still have finite diversity, we ask: how many samples are necessary to get good performance on a task? Our main findings are:

1. Contrastive learning algorithms can be naturally extended to learning from generative samples, where different "views" of the data are created via transformations in the model's latent space.

---

[1]E.g., see https://github.com/justinpinkney/awesome-pretrained-stylegan, hosting many pretrained models, without datasets.

2. These latent-space transformations can be combined with standard data transformations ("augmentations") in pixel-space to achieve better performance than either method alone.

3. To generate positive training pairs in latent space, simple Gaussian transformations work as well as more complicated latent-space transformations, with the optimal standard deviation being not too large and not too small, obeying an inverse-U-shaped curve like that observed for contrastive learning from real data (Tian et al., 2020).

4. Generative models can potentially produce an unbounded number of samples to train on; we find that performance improves by training on more samples, but sub-logarithmically.

5. Learning representations from generative models can outperform learning directly from the data that trained those models, provided that the generative model is of sufficiently quality (we observe this result on StyleGAN2 (Karras et al., 2020) fit to cars but not on BigBiGAN (Donahue & Simonyan, 2019) or BigGAN (Brock et al., 2019) fit to ImageNet1000 (Deng et al., 2009)).

## 2 RELATED WORK

**Learning from Synthetic Data.** Using synthetic data has been a prominent method for learning in different domains of science and engineering, with different goals including privacy-preservation and alternative sample generation (Sakshaug & Raghunathan, 2010; Tucker et al., 2020; Nußberger et al., 2020; Khan et al., 2019; Dan et al., 2020). In computer vision, synthetic data has been extensively used as a source for training models, for example in semantic segmentation (Chen et al., 2019; Ros et al., 2016), human pose estimation (Varol et al., 2017; Ionescu et al., 2013; Shakhnarovich et al., 2003), optical flow (Mayer et al., 2016; Fischer et al., 2015), or self-supervised multitask learning (Ren & Lee, 2018). In most prior work, the synthetic data comes from a traditional simulation pipeline, e.g., via rendering a 3D world with a graphics engine, or recently a noise generator (Baradad et al., 2021). We instead study the setting where the synthetic data is sampled from a deep generative model.

Recent works have used generative networks, such as GANs (Goodfellow et al., 2014), to improve images generated by graphics engines, closing the domain gap with real images (Shrivastava et al., 2017; Hoffman et al., 2018). Ravuri & Vinyals (2019) show that even though there is a gap between GAN-generated images and real ones, mixing the two sources can lead to improvements over models trained purely on real data. GAN-generated images are also used to learn inverse graphics. Zhang et al. (2021a) use StyleGAN to generate multiple views of a given object, allowing to extract 3D knowledge that can be mapped back into the latent space.

Highly related to our paper is the work of Besnier et al. (2020), which uses GAN-generated images to train a classifier. While they focus on using a class-conditional generator to train a classifier of those same classes, our work targets the more general setting of visual representation learning, and we mainly focus on the "unsupervised" setting where an unconditional generative model produces unlabeled samples to learn from.

Recent works (Yang et al., 2021; Jeong & Shin, 2021) have studied combining contrastive and adversarial losses in GAN training, showing benefits in both image synthesis and representation learning. In contrast to our work, these approaches require access to real data during training.

Recent work has also explored using StyleGAN (Karras et al., 2019) to generate training data for semantic segmentation (Zhang et al., 2021b; Tritrong et al., 2021). These approaches differ from ours in that they use intermediate layers of the generator network as their representation of images, whereas our method does not require internal access to the generator; instead we simply treat the generator as a black-box source of training data for a downstream representation learner.

GAN-based data augmentation has also been used, in conjunction with real data, to improve data efficiency for GAN training (Yang et al., 2021) as well as robustness at training time (Mao et al., 2020) and test-time (Chai et al., 2021). In contrast to these methods, we explore representation learning without access to the real data at all, relying only on samples from a black-box generative model to train our systems.

**Contrastive Representation Learning.** Contrastive learning methods (van den Oord et al., 2019; Wu et al., 2018; Hénaff et al., 2019; Tian et al., 2019; He et al., 2020; Chen et al., 2020c) have greatly advanced the state of the art of self-supervised representation learning. The idea of contrastive learning is to contrast positive pairs with negative pairs (Hadsell et al., 2006). Such pairs can be easily constructed with various data formats, and examples include different augmentations

or transformations of images (Chen et al., 2020c), cross-modality alignment (Tian et al., 2019; Morgado et al., 2020; Patrick et al., 2020), and graph structured data (Hassani & Khasahmadi, 2020). One key ingredient for the success of contrastive learning is well-chosen data transformations (Chen et al., 2020c; Tian et al., 2020; Xiao et al., 2020). While all these approaches conduct data transformations, or "augmentations", in the raw pixel space, in this paper we explore the possibility of transforming training points in the latent space of a GAN.

**Generative Representation Learning.** Generative models learn representations by modeling the data distribution $p(\mathbf{x})$. In VAEs (Kingma & Welling, 2013), each data point is encoded into a latent distribution, from which codes are sampled to reconstruct the input by maximizing the data likelihood. GANs (Goodfellow et al., 2014) model data generation through a minimax game, after which the discriminator can serve as a good representation extractor (Radford et al., 2015). In ALI (Dumoulin et al., 2016), BiGAN (Donahue et al., 2017), and BigBiGAN (Donahue & Simonyan, 2019) both image encoding and latent decoding are modeled simultaneously, and the encoder turns out to be a representation learner. Similar to the trend in natural language processing (Devlin et al., 2018; Radford et al.), autoregressive models (Van den Oord et al., 2016; Chen et al., 2020b) have been adopted to learn representations from raw pixels. These prior works show ways to *jointly* train a representation alongside a generative model, using a training set of real data for learning. Our exploration qualitatively differs in that we assume we are *given* a black-box generative model, and *no real training data*, and the goal is to learn an effective representation by sampling from the model.

## 3 METHOD

Standard representation learning algorithms take a *dataset* $\{\mathbf{x}_i\}_{i=1}^N$ as input, and produce an encoder $F : \mathbf{x} \to \mathbf{e}$ as output, where $\mathbf{e}$ is a vector representation of $\mathbf{x}$. Our method, in contrast, takes a *generative model* $G$ as input, in order to produce $F$ as output. We restrict our attention to implicit generative models (IGMs) (Mohamed & Lakshminarayanan, 2016), which map from latent variables $\mathbf{z}$ to sampled images $\mathbf{x}$, i.e. $G : \mathbf{z} \to \mathbf{x}$. Many currently popular generative models have this form, including GANs (Goodfellow et al., 2014), VAEs (Kingma & Welling, 2013), and normalizing flows (Rezende & Mohamed, 2015). We also consider class-conditional variants, which we denote as $G_y : \mathbf{z}, y \to \mathbf{x}$, where $y$ is a discrete class label. In our experiments, we only investigate GANs, but note that the method is general to any IGM with latent variables.

To learn $F$, from either real data or model samples, we can pick any of a large variety of representation learners: autoencoders (Ballard, 1987), rotation prediction (Gidaris et al., 2018), colorization (Zhang et al., 2016), etc. We focus on contrastive methods due to their strong performance and natural extension to using latent transformations to define positive pairs. These methods are illustrated in Fig. 3 in the first and the second rows. We also examine the effectiveness of non-contrastive methods, namely an inverter (similar to the encoder in the autoencoders) as illustrated in the third row of Fig. 3. For conditional IGMs, which generate data from class labels, we also consider using a label-classifier as the representation learner.

In the following sections, we first define the contrastive framework with different sampling strategies for creating views via pixel transformations and latent transformations. We then define the non-contrastive frameworks.

### 3.1 CONTRASTIVE LEARNING FRAMEWORK

Modern contrastive methods learn embeddings that are invariant to certain nuisance transformations of the data, or different "viewing" conditions. Two "views" of the same scene are pulled together in embedding space while views of different scenes are pushed apart. A common objective is the InfoNCE (van den Oord et al., 2019). We use the following variant of this objective:

$$\mathcal{L}_{\text{NCE}} = -\mathbb{E}\left[\log \frac{e^{\tau F(\mathbf{x}_a)^T F(\mathbf{x}_p)}}{e^{\tau F(\mathbf{x}_a)^T F(\mathbf{x}_p)} + \sum_{k=1}^{K} e^{\tau F(\mathbf{x}_a)^T F(\mathbf{x}_n^k)}}\right] \tag{1}$$

Here $\mathbf{x}_a$ is an *anchor* image, $\{\mathbf{x}_a, \mathbf{x}_p\}$ is a *positive* pair of images, i.e. a pair we want to bring together in embedding space. $\{\mathbf{x}_a, \mathbf{x}_n^k\}$ is a *negative* pair of images, which we want to push apart.

Different contrastive learning algorithms differ in how the positive and negative pairs are defined. Commonly, $\mathbf{x}_a$ and $\mathbf{x}_p$ are two different transformations (i.e. data augmentations in pixel space) of

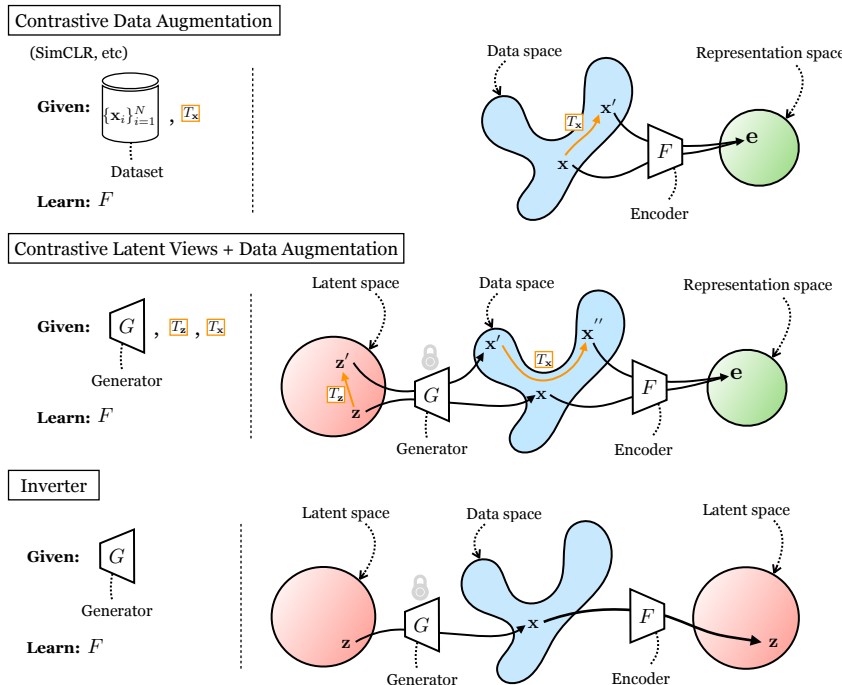

Figure 3: Three different methods for learning representations. The first row illustrates a standard contrastive learning framework (e.g., SimCLR (Chen et al., 2020c)) in which positive pairs as sampled as transformations of training images $\mathbf{x}$. The second and third rows show the new setting we consider: we are given a generator, rather than a dataset, and can use the latent space (input) of the generator to control the production of effective training data. $T_{\mathbf{x}}$ refers to transformations applied in pixel-space and $T_{\mathbf{z}}$ denotes transformations in latent-space. The second row illustrates a contrastive learning approach in this setting and the third row shows an approach that simply inverts the generative model. For contrastive learning, negatives are omitted for clarity.

the same underlying image, and $\mathbf{x}_n^k$ is a transformation of a different randomly selected image (Chen et al., 2020c; He et al., 2020; Gidaris et al., 2018).

Our analysis focuses both on how the datapoints are sampled in the first place, and on how they are transformed to create negative and positive pairs:

1. What happens when $\mathbf{x}$ is *fake* data sampled from an IGM rather than *real* data from a dataset?
2. How can we use transformations in pixel space and in latent space to define positives/negatives?

Different answers to these two questions yield the specific methods we compare, as described in the next sections.

## 3.2 Sampling positive and negative pairs

We first describe several schemes for sampling from both real and generated datapoints, with transformations applied in either latent space or pixel space.

### 3.2.1 Contrastive pixel transformations (i.e. SimCLR)

If we are given a dataset, and a set of pixel-space transformations, $T_{\mathbf{x}}$, that we wish to be invariant to, a standard approach is to use the following sampling strategy:

$$\hat{\mathbf{x}}_a, \hat{\mathbf{x}}_n^k \sim D \qquad \triangleleft \text{ sample anchor and negatives} \qquad (2)$$

$$\mathbf{x}_a, \mathbf{x}_p \sim T_{\mathbf{x}}(\hat{\mathbf{x}}_a) \qquad \triangleleft \text{ generate positive pair} \qquad (3)$$

$$\mathbf{x}_n^k \sim T_{\mathbf{x}}(\hat{\mathbf{x}}_n^k) \qquad \triangleleft \text{ generate negatives} \qquad (4)$$

where $\sim D$ refers to drawing a random image uniformly from the dataset $D = \{\mathbf{x}_i\}_{i=1}^N$. This setting is depicted in the top row of Fig. 3. We use SimCLR (Chen et al., 2020c) as an instantiation of this approach.

### 3.2.2 CONTRASTIVE LATENT VIEWS + PIXEL TRANSFORMATIONS

If we are given an unconditional IGM $G$, we may also define a set of *latent* transformations, $T_{\mathbf{z}}$, that we wish our representation to be invariant to. We can use this method with or without pixel-space transformations $T_{\mathbf{x}}$ applied as well. With both $T_{\mathbf{z}}$ and $T_{\mathbf{x}}$ applied, we have the following fake data generating process:

$$
\begin{aligned}
&\mathbf{z}_a, \mathbf{z}_n^k \sim p_{\mathbf{z}} && \triangleleft \text{sample latents} && (5) \\
&\mathbf{x}_a \sim T_{\mathbf{x}}(G(\mathbf{z}_a)) && \triangleleft \text{generate anchor sample} && (6) \\
&\mathbf{x}_p \sim T_{\mathbf{x}}(G(T_{\mathbf{z}}(\mathbf{z}_a))) && \triangleleft \text{generate positive sample} && (7) \\
&\mathbf{x}_n^k \sim T_{\mathbf{x}}(G(T_{\mathbf{z}}(\mathbf{z}_n^k))) && \triangleleft \text{generate negatives} && (8)
\end{aligned}
$$

In practice we set $p_{\mathbf{z}}$ to be the truncated normal distribution, $\mathcal{N}^t(\mu, \sigma, t)$, with truncation $t$ (Brock et al., 2019). This scenario is depicted in the middle row of Fig. 3. Refer to App. C.1.1 for the class-conditional formulation.

## 3.3 CREATING VIEWS WITH $T_{\mathbf{x}}$ AND $T_{\mathbf{z}}$

We refer to transformations as "views" of the data (either real or fake). Figure 4 shows different views generated according to all the following methods. For pixel-space transformations, $T_{\mathbf{x}}$, many options have been previously proposed, and in our experiments we choose the transformations from SimCLR (Chen et al., 2020c) as they are currently standard and effective. Our framework could be updated with better transformations as they are developed in future work.

Our work is the first, to our knowledge, to explore latent-space transformations, $T_{\mathbf{z}}$, for contrastive representation learning. Therefore, we focus our analysis on studying the properties of $T_{\mathbf{z}}$ and explore the following options:

### 3.3.1 GAUSSIAN LATENT VIEWS

Many IGMs have the remarkable property that nearby points in latent space map to semantically related generated images (Jahanian et al., 2020). This suggests that we can define positive views as nearby latent vectors (refer to Sec. 4.1.2 for an empirical proof). A simple way to do so is to define the latent transformation to just be a small offset applied to the sampled $\mathbf{z}$ vector. We use truncated Gaussian offsets $\mathbf{w}_{\texttt{Gauss}}$ as a simple instantiation of this idea:

$$
\mathbf{w}_{\texttt{Gauss}} \sim \mathcal{N}^t(\mu, \sigma, t) \tag{9}
$$

$$
T_{\mathbf{z}}(\mathbf{z}) = \mathbf{z} + \mathbf{w}_{\texttt{Gauss}} \tag{10}
$$

where $\mathcal{N}^t(\mu, \sigma, t)$ is the truncated Normal distribution with truncation $t$ (Brock et al., 2019).

| (a) Original | (b) Gaussian Views | (c) Steered Views | (d) SimCLR Views | (e) Gaussian + SimCLR Views | (f) Steer + SimCLR Views |

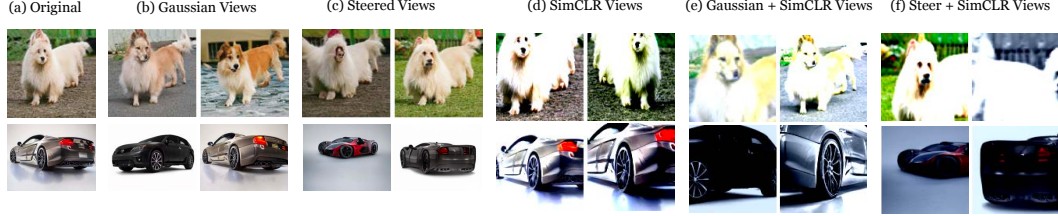

Figure 4: Examples of different transformation methods for unconditional IGM data. Top row shows samples of BigBiGAN trained on ImageNet1000, and the bottom row shows samples from the StyleGAN2 LSUN CAR.

### 3.3.2 STEERED LATENT VIEWS

Can we find latent transformations that are more directly related to semantics? We leverage the recent method of (Jahanian et al., 2020), which finds latent vectors $\mathbf{w}_{\texttt{steer}}$ that achieve target effects in image space, such as shifting an image up and down or changing it is brightness. That is, $T_{\mathbf{z}}(\mathbf{z}) = \mathbf{z} + \mathbf{w}_{\texttt{steer}}$, where $\mathbf{w}_{\texttt{steer}}$ is a latent vector trained to match a target pixel-space transformation $T$ according to the objective $\mathbf{w}_{\texttt{steer}} = \arg\min_{\mathbf{w}} \mathbb{E}_{\mathbf{z}, \alpha}[||G(\mathbf{z} + \alpha \mathbf{w}) - T(G(\mathbf{z}), \alpha)||]$. For example, $T$ could be "shift left", parameterized by the strength of the transformation $\alpha$, e.g. the number of pixels to shift. In App. B we describe the exact steering transformations in more detail.

### 3.4 NON-CONTRASTIVE ALTERNATIVES

We can also learn image representations without a contrastive loss. Here we describe one possible approach, an inverter. Appendix C.2.3 describes another non-contrastive approach, a classifier, that only applies in the class-conditional IGM setting.

**Inverter** The $\mathbf{z}$ vector that generates an image can itself be considered a vector representation of that image. This suggests that simply inverting the generator may result in a useful embedding function. For a given IGM $G$, we learn an encoder $F$ that inverts G by minimizing $\|F(G(\mathbf{z})) - \mathbf{z}\|_2^2$.

## 4 EXPERIMENTS

To study the methods in Sec. 3, we analyze the behaviour and effectiveness of the representations learned from unconditional IGMs here and, in the interest of space, report the results on class-conditional IGMs in App. C. We experiment on two unconditional IGMs: the generator from Big-BiGAN (Donahue & Simonyan, 2019) and the StyleGAN2 LSUN CAR generator (Karras et al., 2020). Note, we do *not* use the encoder or training scheme from BigBiGAN, we simply use its learned generator G as an off-the-shelf unconditional ImageNet generator.

For ImageNet representation learning, we investigate two settings in our experiments: 1) training and evaluating representations on data at the scale of ImageNet1000 (Deng et al., 2009), and 2) a lighter weight protocol where we train and evaluate representations only on data at the scale of ImageNet100 (Tian et al., 2019) (a 100-class subset of ImageNet1000). In the ImageNet1000 setting, the real data encoders are trained on the ImageNet1000 dataset, the IGM encoders are trained on $1.3M$ anchor images (which roughly matches the size of ImageNet1000). We use SGD with learning rate of 0.03, batch size of 256, and 20 epochs. In the ImagetNet100 setting, the real data encoders are trained on ImageNet100, the unconditional IGM encoders are trained on $130K$ anchor images sampled unconditionally (note that the unconditional model implicitly is still sampling from all 1000 classes since the generative model itself was fit to ImageNet1000). We use SGD with learning rate of 0.03, batch size of 256, for 200 epochs. In all settings, we use $128 \times 128$ images, leading to lower than state-of-the-art performance on the real datasets, more details in App. D.

We evaluate the ImageNet learned representations by training a linear classifier on top of the learned embeddings, for either ImageNet1000 or ImageNet100, matching the setting in which the encoder was trained, and report Top-1 class accuracy. For both ImageNet1000 and ImageNet100 settings, we use SGD with batch size of 256 over 60 epochs, and learning rates of 0.3 for real and 2 for IGMs, using a cosine decay schedule. We also evaluate on the Pascal VOC2007 dataset (Everingham et al., 2010) as a held out data setting, selecting hyper-parameters via cross-validation, following (Kornblith et al., 2019) and report the mean average precision ($AP$) measures. See App. E for results on the object detection task.

For the StyleGAN2 LSUN CAR experiments, we use the pretrained model on $893K$ images from the LSUN CAR dataset. See App. A for obtaining the real data. For the contrastive model trained on this data, we use SGD with learning rate of 0.03, batch size of 256, and 20 epochs. We evaluate the learned representations by training a linear classifier on top of the learned embeddings, for either ImageNet1000 or Stanford Car classification task (Krause et al., 2013) (196 car models with roughly $8K$ train and $8K$ val), and report Top-1 class accuracy. For both experiments, we use SGD with batch size of 256 over 100 epochs, and learning rates of 2, using a cosine decay schedule.

### 4.1 CONTRASTIVE LEARNING METHODS

To generate datasets containing multiple views produced by latent transformations, we first sample $1.3M$ anchor view images in the ImageNet1000 setting (described above). All the anchors are generated with $\mathbf{z} \sim \mathcal{N}(0, 1, 2)$. We then generate one neighbor view for each anchor view by following different strategies for view creation, Gaussian and steer, described in Sec. 3.3. For the Gaussian view, we tune the standard deviation of $\mathbf{w}_{\texttt{Gauss}}$ to be 0.2 for our unconditional IGMs (i.e. $\mathbf{w}_{\texttt{Gauss}} \sim \mathcal{N}(0, 0.2, 2)$), and we use this setting in all of our experiments. For steering, we learn the latent walk $\mathbf{w}_{\texttt{steer}}$ as the summation of individual walks with randomly sampled steps for different camera transformations, i.e. horizontal and vertical shifts, zoom, 2D and 3D rotations as well as color transformations. The details for training $\mathbf{w}_{\texttt{steer}}$ follows (Jahanian et al., 2020), and we report them in App. B. We use similar view creation strategies for Stylegan2 LSUN CAR, i.e., each anchor is generated with $\mathbf{z} \sim \mathcal{N}(0, 1, 0.9)$, and per anchor image, a neighbor is generated with the same steering or Gaussian strategy with $std = 0.25$ apart from its anchor in the case of the Gaussian

views. The resulting images of our latent view creations are shown in Fig. 4(b,c). When needed, we further combine the latent views with pixel transformations, as shown in Fig. 4(e,f). See App. A for details on how we handle cropping. In the following sections, we learn visual representations of the generated images by training a ResNet-50 via SimCLR (Chen et al., 2020c) for the datasets generated from unconditional IGMs.

### 4.1.1 EFFECT OF THE LATENT AND IMAGE TRANSFORMATIONS

First, we experiment with the effect of using pixel transformations on images sampled by the IGMs, without latent transformations, i.e. $T_{\mathbf{z}} = -$. Next, we enable both latent and pixel transformations together, where the latent transformation is either Gaussian or steer, i.e. $\mathbf{z} + \mathbf{w}_{\texttt{Gauss}}$ or $\mathbf{z} + \mathbf{w}_{\texttt{steer}}$, respectively. We report in Table 1 and Table 2 the transfer results on unconditional IGMs, for the described transformations. From this table we observe similar trends across the unconditional IGMs: while pixel transformations $T_{\mathbf{x}}$ lead to strong representations, these are significantly improved by adding latent transformations $T_{\mathbf{z}}$. Comparing different types of latent transformations, we find that

| Training Method | | | | Transfer Task | |
| --- | --- | --- | --- | --- | --- |
| Data distribution | $T_{\mathbf{z}}$ | $T_{\mathbf{x}}$ | Objective | ImageNet1000 Top-1 Accuracy | VOC07 Classification AP |
| Real | − | SimCLR Augs. | Contrastive | 43.90 | 0.67 |
| Generated | − | SimCLR Augs. | Contrastive | 35.69 | 0.57 |
| Generated | $\mathbf{z} + \mathbf{w}_{\texttt{Gauss}}$ | − | Contrastive | 28.88 | 0.52 |
| Generated | $\mathbf{z} + \mathbf{w}_{\texttt{Gauss}}$ | SimCLR Augs. | Contrastive | **42.58** | **0.64** |
| Generated | $\mathbf{z} + \mathbf{w}_{\texttt{steer}}$ | − | Contrastive | 26.52 | 0.49 |
| Generated | $\mathbf{z} + \mathbf{w}_{\texttt{steer}}$ | SimCLR Augs. | Contrastive | 41.78 | 0.63 |
| Generated | − | − | Inverter | 26.43 | 0.49 |

Table 1: Results on unconditional IGMs where real data is sampled from ImageNet1000 and is distributed as $\mathbf{x} \sim T_{\mathbf{x}}(D)$, and generated data is sampled from BigBiGAN and distributed as $\mathbf{x} \sim T_{\mathbf{x}}(G(T_{\mathbf{z}}(\mathbf{z})))$. $T_{\mathbf{z}} = -$ indicates that no transformation is applied. For the *Contrastive* objective, positive and negative views are defined as described in Sec. 3.2.1 (without using class labels).

| Training Method | | | | Transfer Task (Top-1 Accuracy) | |
| --- | --- | --- | --- | --- | --- |
| Data distribution | $T_{\mathbf{z}}$ | $T_{\mathbf{x}}$ | Objective | ImageNet1000 | Stanford Car Classification |
| Real | − | SimCLR Augs. | Contrastive | 30.30 | 39.68 |
| Generated | − | SimCLR Augs. | Contrastive | 28.33 | 40.89 |
| Generated | $\mathbf{z} + \mathbf{w}_{\texttt{Gauss}}$ | − | Contrastive | 24.50 | 24.50 |
| Generated | $\mathbf{z} + \mathbf{w}_{\texttt{Gauss}}$ | SimCLR Augs. | Contrastive | **30.06** | **49.79** |

Table 2: Results on unconditional IGMs where generated data is sampled from StyleGAN2 LSUN CAR and distributed as $\mathbf{x} \sim T_{\mathbf{x}}(G(T_{\mathbf{z}}(\mathbf{z})))$. $T_{\mathbf{z}} = -$ indicates that no transformation is applied. For the *Contrastive* objective, positive and negative views are defined as described in Sec. 3.2.1 (without using class labels).

Gaussian transformations work best. This is a somewhat surprising result. Despite designing views through steering methods, transforming randomly in all latent directions provides similar or better performance.

Another important observation is that training on StyleGAN2 fit to LSUN CAR outperforms training directly on LSUN CAR (Table 2). This result demonstrates that generative data can, in some cases, be more useful for representation learning than training directly on real photographs. The success here may be due to the fact that the StyleGAN2 samples are nearly identical to real photos in their realism and diversity (FID = 2.32). Therefore we might expect StyleGAN2 samples, trained only with $T_{\mathbf{x}}$, to more or less match the performance of training on real data with $T_{\mathbf{x}}$, and indeed this is what we find (Table 2; compare first two rows). It is then unsurprising that adding $T_{\mathbf{z}}$, which can only be done for generative data, can boost performance further.

### 4.1.2 LIMITS OF THE LATENT TRANSFORMATIONS

Prior work on contrastive learning has found that there is an optimal point in how much two views share information (Tian et al., 2020). Similarly, in our Gaussian latent view creation, we may expect there to be a sweet spot for how far we can go when sampling a neighbor view relative to an anchor view. To study that sweet spot for the BigBiGAN generator, we vary the standard deviation of the Gaussian transformation. For this experiment we use the ImageNet100 setting. Fig. 5 illustrates the results: linear transfer performance exhibits an inverse-U-shaped curve with respect to Gaussian standard deviation, peaking at $std = 0.2$.

### 4.2 NON-CONTRASTIVE METHODS

**Inverter** We experiment with an inverter, as illustrated in Fig. 3. We learn an image encoder by minimizing the mean squared error between the original and predicted latent code $\mathbf{z}$. Similar to the

contrastive setting, we train a ResNet-50 encoder and replace the last layer with a fully connected layer to predict the latent vector $\mathbf{z}$ of a $128 \times 128$ image. We train the encoder with a dataset of $1.3M$ images and their latent vectors, i.e. pairs of $\langle G(\mathbf{z}), \mathbf{z} \rangle$. Note we don't apply image transformations on $G(\mathbf{z})$ given that directions in $\mathbf{z}$-space encode basic transformations like shifting and color change (Jahanian et al., 2020) so $\mathbf{z}$ is not invariant to these transformations. The results of using the unconditional IGM representations in Table 1 suggest this non-contrastive approach performs poorly in comparison with the contrastive methods.

**Comparison to BigBiGAN's *encoder*** We finally compare the pretrained BigBiGAN encoder (Donahue & Simonyan, 2019) (a non-contrastive approach) with our proposed methods. Note the BigBiGAN encoder is jointly trained with the generator, thus requiring access to the original real dataset during training. This is *not* the setting we are targeting but we provide this comparison nonetheless to see where our methods stand compared to prior work that involved a generative model in learning a representation. To match the setup with our experiments, we take the BigBiGAN ResNet50 encoder after average pooling (without the head) and use it in our linear classification of ImageNet100 test. Further, note that this encoder is trained on ImageNet1000 and accepts as input images of size $256 \times 256$, whereas for this experiment we compare against our encoders trained on the ImageNet100 training sets and use images of size $128 \times 128$. The top-1 accuracy from the Big-BiGAN encoder is $55.70\%$ which is better than our inverter but still worse than all the contrastive methods trained and tested on ImageNet100 (see the numbers in in Table 4 in App. E).

### 4.3 Effect of the number of samples

In the synthetic data setting, we can create infinite data (albeit with finite diversity). A natural question arises: how many samples do we need for good coverage and good visual representations? To answer this question, we run experiments in the ImageNet100 setting and evaluate the linear classification performance for Gaussian views combined with pixel transformations, as we vary the number of unique samples. We show the results in Fig. 6. Note that we train all the encoders for the same number of iterations, equivalent to 200 epochs on ImageNet100 (matching the experiments shown in table 4 in App. E). This means, as the number of images (x-axis) increases, the number of times the model revisits a seen image decreases. As Fig. 6 illustrates, the performance increases with more samples, but sub-logarithmically. These findings are consistent with recent work that found a small gap in generalization performance between online learning (infinite data) and a sufficiently large offline regime (Nakkiran et al., 2020).

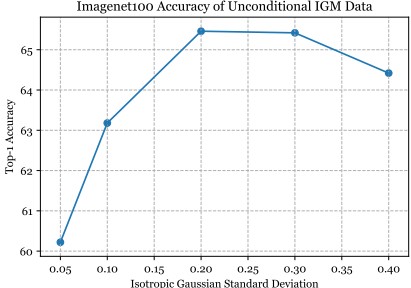
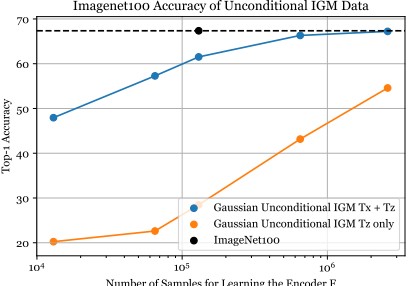

Figure 5: Effect of the distance between latent views on contrastive learning. We vary the standard deviation of a Gaussian $T_{\mathbf{z}} = \mathbf{z} + \mathbf{w}_{\text{Gauss}}$ and measure linear transfer to ImageNet100.

Figure 6: Effect of the number of samples for training the representation learner, evaluated using linear transfer to ImageNet100. "Gaussian" refers to the Gaussian views ($T_{\mathbf{z}} = \mathbf{z} + \mathbf{w}_{\text{Gauss}}$).

## 5 Conclusion

We investigated how to learn visual representations from IGMs (implicit generative models), when they are given as a black-box and without any access to their training data. IGMs make it possible to generate multiple views of similar image content by sampling nearby points in the latent space of the IGM. These views can be used for contrastive learning or as input to other representation learning algorithms that associate multiple views of the same scene. Our results demonstrate that leveraging latent-space views can improve performance beyond learning from pixel-space transformations alone. Representations learned from large quantities of generative data rival and sometimes outperform the transfer performance of representations learned from the real datasets. As generative models improve, we expect that training vision systems on generative data may become an increasingly important tool in our toolkit.

## 6 ETHICS STATEMENT

This work studies how to learn useful image representations given data generated from IGMs as opposed to real data. This framework can provide several societal advantages currently faced in real datasets, including protecting the privacy and usage rights of real images (Tucker et al., 2020; Du-Mont Schütte et al., 2021), removing sensitive attributes (Liao et al., 2019), or reducing biases (Tan et al., 2020; Ramaswamy et al., 2020) and therefore learning fairer representations.

At the same time, learning representations from IGMs brings several challenges and risks, that need to be addressed. Generative models can in some cases reveal the data they are trained on Hayes et al. (2018); Chen et al. (2020a), posing risks for privacy preservation. They can also amplify biases in datasets (Jain et al., 2020; Menon et al., 2020), which can lead to negative societal impacts if they are not audited properly (Raji et al., 2020a;b; Mitchell et al., 2019), or they are used in the wrong contexts Buolamwini & Gebru (2018). Making good use of the learned representations will require addressing the above issues by studying mitigation strategies, as well as methods to audit implicit generative models.

ACKNOWLEDGMENTS

Author A.J. thanks Kamal Youcef-Toumi, Boris Katz, and Antonio Torralba for their support. We thank Antonio Torralba, Janne Hellsten, David Bau, and Tongzhou Wang for helpful discussions.

This research was supported in part by IBM through the MIT-IBM Watson AI Lab. The research was also partly sponsored by the United States Air Force Research Laboratory and the United States Air Force Artificial Intelligence Accelerator and was accomplished under Cooperative Agreement Number FA8750-19-2-1000. The views and conclusions contained in this document are those of the authors and should not be interpreted as representing the official policies, either expressed or implied, of the United States Air Force or the U.S. Government. The U.S. Government is authorized to reproduce and distribute reprints for Government purposes notwithstanding any copyright notation herein.

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

# APPENDIX

## A  CROPPING DETAILS

For all methods that use only $T_\mathbf{z}$, we generate images at the native resolution of the generator output ($128 \times 128$ for BigBiGAN, $256 \times 256$ for BigGAN, and $512 \times 512$ for StyleGAN) then take a center crop to produce the $128 \times 128$ pixel inputs to our models. In preliminary experiments we found that this strategy worked better than downsampling the generated images to $128 \times 128$. Note that in our figures for $T_\mathbf{z}$ we visualize the generated images before this crop is applied.

For StyleGAN LSUN CAR, the dataset consists of rectangular images padded with black borders. For all methods we remove the padding before training on these images. This results in non-square images. Methods that use $T_\mathbf{x}$ take a random square crop (one of the SimCLR augmentations) and for methods without $T_\mathbf{z}$ we take a $128 \times 128$ pixel center square crop.

Furthermore, for StyleGAN LSUN CAR, to compare against training on real data, we need these $893K$ images; as they are not publicly released, we recreate this dataset by using the LSUN CAR dataset sampling code from the StyleGAN2 repository[2].

## B  CREATING STEERING VIEWS

For creating steering views, we apply the walk $\mathbf{w}_\texttt{steer}$ in Sec. 3.3.2. Here we expand this vector as the summation of individual walks with randomly sampled steps for different camera transformations, i.e. horizontal and vertical shifts, zoom, 2D and 3D rotations as well as color transformations. In fact, the final walk is:

$$\mathbf{w}_\texttt{steer} = \sum_i \alpha_i \mathbf{w}^i_\texttt{steer}, \tag{11}$$

for $i \in \{\texttt{shiftx}, \texttt{shifty}, \texttt{zoom}, \texttt{rot2D}, \texttt{rot3D}, \texttt{color}\}$, each representing a pixel-space transformation. That is:

$$\mathbf{w}^i_\texttt{steer} = \arg\min_\mathbf{w} \mathbb{E}_{\mathbf{z},\alpha}[||G(\mathbf{z}+\alpha\mathbf{w}) - T_i(G(\mathbf{z}),\alpha)||], \tag{12}$$

where $T_i$ can be "shift left/right", parameterized by the strength of the transformation $\alpha$, e.g. the number of pixels to shift.

Following (Jahanian et al., 2020), coefficient $\alpha$ determines what magnitude we want for each transformation. For example, for $\texttt{color}$, we add $\alpha$ to each RGB channel of an image (and have three $\mathbf{w}$ vectors in the latent space). Similarly, for each $i \in \{\texttt{shiftx}, \texttt{shifty}, \texttt{rot2D}, \texttt{rot3D}\}$, $\alpha$ determines how many pixels we shift or rotate the image. For $\texttt{zoom}$, we use $\alpha$ to determine how much scale we should apply to the image. For that reason, we use $log(\alpha)$ in the latent space but $\alpha$ in the pixel space (to define the edits).

In order to train $\mathbf{w}_\texttt{steer}$, we create a dataset of target images and optimize for the best $\mathbf{w}^i_\texttt{steer}$ walks that in a linear summation yield images close to the target images. Here, each target image is a composition of all the pixel-space transformations applied to the $G(\mathbf{z})$ image with random magnitudes ($\alpha$ for each $T_i$).

After learning the $\mathbf{w}^i_\texttt{steer}$ walks together, we can randomly draw a $\mathbf{z}$ vector and add them it to (with random steps $\alpha_i$), and this results in a transformed image. This transformed image will serve as a positive view for the image of the given $\mathbf{z}$ vector.

## C  CLASS-CONDITIONAL IGMS

### C.1  METHOD

Given a class-conditional IGM, we can do the same formulations as in Sec. 3.2.2 except conditioned on class labels, i.e. we simply replace $G(\cdot)$ with $G(\cdot, y)$, using $y = y_p \sim \texttt{Cat}(M)$ for the anchor and positive sample, and an independent draw $y = y_n \sim \texttt{Cat}(M)$ for the negative sample. Additionally,

---

[2]https://github.com/NVlabs/stylegan2-ada-pytorch

taking inspiration from SupCon, in each batch, one positive is set with the same $\mathbf{z}$ as the anchor (but different pixel space transformation) while the other positives in the batch are generated from independently drawn $\mathbf{z} \sim p_{\mathbf{z}}$.

### C.1.1 SUPERVISED CONTRASTIVE PIXEL TRANSFORMATIONS (I.E. SUPCON)

Given a *labeled* dataset $D = \{\mathbf{x}_i, y_i\}_{i=1}^N$, with $M$ classes, we can leverage the labels to define positives as images that share the same labels and negatives as randomly sampled images from other classes:

$$y_p \sim \mathtt{Cat}(M) \qquad \triangleleft \text{sample positive class} \qquad (13)$$

$$\hat{\mathbf{x}}_a, \hat{\mathbf{x}}_p \sim D_{y_p}, \hat{\mathbf{x}}_n^k \sim D_{/y_p} \qquad \triangleleft \text{sample pairs} \qquad (14)$$

$$\mathbf{x}_a \sim T_{\mathbf{x}}(\hat{\mathbf{x}}_a), \mathbf{x}_p \sim T_{\mathbf{x}}(\hat{\mathbf{x}}_p) \qquad \triangleleft \text{generate positive} \qquad (15)$$

$$\mathbf{x}_n^k \sim T_{\mathbf{x}}(\hat{\mathbf{x}}_n^k) \qquad \triangleleft \text{generate negatives} \qquad (16)$$

where $\mathtt{Cat}$ is the categorical distribution and $D_{y_p} = \{x_i \in D | y_i = y_p\}$, $D_{/y_p} = \{x_i \in D | y_i \neq y_p\}$. Following SupCon (Khosla et al., 2020), in each batch one of the positives is specially set as $\mathbf{x}_p \sim T_{\mathbf{x}}(\hat{\mathbf{x}}_a)$.

### C.1.2 INDEPENDENT LATENT VIEWS

The simplest method is:

$$T_{\mathbf{z}}(\cdot) = p_{\mathbf{z}} \qquad (17)$$

That is, the transformation simply produces a new random draw from $p_{\mathbf{z}}$. In the unconditional setting, this can be considered a naive baseline where the two views share no information about the image. However, in the class-conditional case, this strategy is actually quite sensible, and in a sense optimal if the goal is to extract class semantics (Tian et al., 2020): the two positive views are two different images that are independent *except* that they share the same class $y_p$.

### C.2 EXPERIMENTS

For the class-conditional IGM, we use BigGAN (Brock et al., 2019) trained on ImageNet1000 (Deng et al., 2009) with the "deep-256" and truncation= 2.0. We use image size of $128 \times 128$ for all the settings (i.e. scaling down BigGAN images).

Similar to unconditional experiments, for ImageNet representation learning, we investigate two settings in our experiments: 1) training and evaluating representations on data at the scale of ImageNet1000 (Deng et al., 2009), and 2) a lighter weight protocol where we train and evaluate representations only on data at the scale of ImageNet100 (Tian et al., 2019) (a 100 class subset of ImageNet1000). In the ImageNet1000 setting, the real data encoders are trained on the ImageNet1000 dataset, the class-conditional IGM encoders are trained on $1.3M$ anchor images (1300 images conditioned on each of the 1000 ImageNet1000 classes, which roughly matches the size of ImageNet1000). We use SGD with learning rate of 0.03, batch size of 256, and 20 epochs. In the ImagetNet100 setting, the real data encoders are trained on ImageNet100, the class-conditional IGM encoders are trained on $130K$ anchor images (1300 images conditioned on each of the 100 classes in ImageNet100). We use SGD with learning rate of 0.03, batch size of 256, for 200 epochs.

We evaluate the ImageNet learned representations by training a linear classifier on top of the learned embeddings, for either ImageNet1000 or ImageNet100, matching the setting in which the encoder was trained, and report Top-1 class accuracy. For both ImageNet1000 and ImageNet100 settings, we use SGD with batch size of 256 over 60 epochs, and learning rates of 0.3 for real and 2 for IGMs, using a cosine decay schedule. We also evaluate on the Pascal VOC2007 dataset (Everingham et al., 2010) as a held out data setting, selecting hyper-parameters via cross-validation, following (Kornblith et al., 2019) and report the mean average precision ($AP$) measures. See App. E for evaluating on the object detection task.

### C.2.1 CONTRASTIVE LEARNING METHODS

IGMs can create multiple views of images via their latent transformations (Jahanian et al., 2020), making them useful for contrastive multi-view learning. In this section, we study the effectiveness of contrastive methods for learning representations from IGMs.

To generate datasets containing multiple views produced by latent transformations, we first sample $1.3M$ anchor view images in the ImageNet1000 setting (described above). All the anchors are generated with $\mathbf{z} \sim \mathcal{N}(0, 1, 2)$. We then generate one neighbor view for each anchor view by following different strategies for view creation, Gaussian and steer, described in Sec. 3.3. For the Gaussian view, we tune the standard deviation of $\mathbf{w}_{\texttt{Gauss}}$ to be $1.0$ for our class-conditional IGMs (i.e. $\mathbf{w}_{\texttt{Gauss}} \sim \mathcal{N}(0, 1.0, 2)$), and we use this setting in all of our experiments. For steering, we learn the latent walk $\mathbf{w}_{\texttt{steer}}$ as the summation of individual walks with randomly sampled steps for different camera transformations, i.e. horizontal and vertical shifts, zoom, 2D and 3D rotations as well as color transformations. See the details in App. B.

The resulting images of our latent view creation strategies are shown in Fig. 7(b,c). In the class-conditional case, we test independent views, i.e. $T_{\mathbf{z}} = p_{\mathbf{z}}$ and qualitatively show examples in Fig. 7(d). When needed, we further combine the latent views with pixel transformations, as shown in Fig. 7(f).

In the following sections, we learn visual representations of the generated images by training a ResNet-50 via SupCon (Khosla et al., 2020) for the class-conditional IGMs.

| (a) Original | (b) Gaussian Views | (c) Steered Views | (d) Independent Latent Views | (e) SupCon Views | (f) Gaussian + SupCon Views |
|---|---|---|---|---|---|

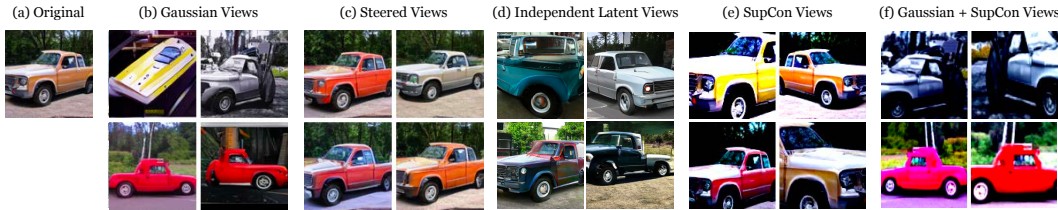

Figure 7: Examples of different transformation methods for **class-conditional** IGMs data. The images are sampled from BigGAN.

### C.2.2 EFFECT OF THE LATENT AND IMAGE TRANSFORMATIONS

First, we experiment with the effect of using pixel transformations on images sampled by the IGMs, without latent transformations, i.e. $T_{\mathbf{z}} = -$. Next, we enable both latent and pixel transformations together, where the latent transformation is either Gaussian or steer, i.e. $\mathbf{z} + \mathbf{w}_{\texttt{Gauss}}$ or $\mathbf{z} + \mathbf{w}_{\texttt{steer}}$, respectively. There is a third latent transformation, independent view, i.e. $T_{\mathbf{z}} = p_{\mathbf{z}}$, only applicable to the class-conditional IGM because of conditioning on the class semantics. We report in Table 3 the transfer results on class-conditional IGMs, respectively, for the described transformations. From this table we observe similar trends across both unconditional and class-conditional IGMs (compare with Table 1. The view generation works well along with the contrastive methods: while pixel transformations $T_{\mathbf{x}}$ lead to strong representations, these are significantly improved by adding latent transformations $T_{\mathbf{z}}$, in the unconditional case. In the class-conditional case, on the other hand, there is only marginal improvement using $T_{\mathbf{z}}$, indicating that the class-label supervision may be sufficient to already lead to strong semantic representations.

### C.2.3 NON-CONTRASTIVE METHODS

We further study representations learned from non-contrastive methods.

**Inverter** We first experiment with an inverter, as illustrated in Fig. 3. We learn an image encoder by minimizing the mean squared error between the original and predicted latent code $\mathbf{z}$. For the class-conditional IGMs, we include an auxiliary cross-entropy loss, to predict the category of the encoded image. For this experiment, we use the ImageNet1000 setting.

Similar to the contrastive setting, we train a ResNet-50 encoder and replace the last layer with a fully connected layer to predict the latent vector $\mathbf{z}$ and the labels $y$ of a $128 \times 128$ image. We

| Training Method | | | | Transfer Task | |
|---|---|---|---|---|---|
| Data distribution | $T_{\mathbf{z}}$ | $T_{\mathbf{x}}$ | Objective | ImageNet1000 Top-1 Accuracy | VOC07 Classification AP |
| Real | − | SimCLR Augs. | Sup. Contrastive | 50.84 | 0.76 |
| Generated | − | SimCLR Augs. | Sup. Contrastive | 48.19 | 0.75 |
| Generated | $\mathbf{z} + \mathbf{w}_{\text{Gauss}}$ | − | Sup. Contrastive | 35.74 | 0.66 |
| Generated | $\mathbf{z} + \mathbf{w}_{\text{Gauss}}$ | SimCLR Augs. | Sup. Contrastive | 46.43 | 0.74 |
| Generated | $\mathbf{z} + \mathbf{w}_{\text{steer}}$ | − | Sup. Contrastive | 36.74 | 0.68 |
| Generated | $\mathbf{z} + \mathbf{w}_{\text{steer}}$ | SimCLR Augs. | Sup. Contrastive | **49.25** | 0.75 |
| Generated | $p_{\mathbf{z}}$ | − | Sup. Contrastive | 36.21 | 0.65 |
| Generated | $p_{\mathbf{z}}$ | SimCLR Augs. | Sup. Contrastive | 48.97 | **0.76** |
| Generated | − | − | Inverter | 31.56 | 0.60 |

Table 3: Results on **class-conditional** IGMs. Real data is sampled from ImageNet1000 and distributed as $\mathbf{x} \sim T_{\mathbf{x}}(D)$. Generated data is sampled from BigGAN and distributed as $\mathbf{x} \sim T_{\mathbf{x}}(G(T_{\mathbf{z}}(\mathbf{z}), y))$. $T_{\mathbf{z}}, T_{\mathbf{x}} = -$ indicates that no transformation is applied. $T_{\mathbf{z}} = p_{\mathbf{z}}$ indicates that the transformation draws a new sample $p_{\mathbf{z}}$, independent of the original $\mathbf{z}$. For the *Sup. Contrastive* objective, positives are defined following App. C.1.1, where two views are treated as positive if and only if they share the same label $y$.

train the encoder with a dataset of $1.3M$ images and their latent vectors, i.e. pairs of $\langle G(\mathbf{z}, y), \mathbf{z} \rangle$. Note we don't apply image transformations on $G(\mathbf{z}, y)$ given that directions in $\mathbf{z}$-space encode basic transformations like shifting and color change (Jahanian et al., 2020) so $\mathbf{z}$ is not invariant to these transformations.

The results of using the the class-conditional IGMs in Table 3 suggest this non-contrastive approach performs poorly in comparison with the contrastive methods.

**Classifier** When learning representations from class-conditional IGMs, we can leverage the labels available and learn representations through a classification objective (softmax cross-entropy). We study the performance of this learning objective in the ImageNet100 setting. We train a classifier with a ResNet-50 backbone to classify images into one of the 100 classes of ImageNet100, applying $T_{\mathbf{x}}$ as data augmentation but not $T_{\mathbf{z}}$. After an embedding has been learned in this way, we evaluate the Top-1 Accuracy on ImageNet100 of a linear classifier on top of the learned representations. This approach achieves 65.2%, performing similarly to the Supervised Contrastive objective, which achieves 66.8% accuracy in the same setting (only $T_{\mathbf{x}}$, evaluation on ImageNet100). Overall, the classifier is a subset of the inverter (which reconstructs both $\mathbf{z}$ and class label $y$), but it allows us to use data pixel augmentations which also helps for higher performance. We provide additional results in App. E on the classifier experiments in Table 6. Note these experiments are in the ImageNet100 setting and can be compared with Table 5.

### C.2.4 EFFECT OF THE NUMBER OF SAMPLES

As discussed in Sec. 4.3, in the synthetic data setting, we can create infinite data (albeit with finite diversity). A natural question arises: how many samples do we need for good coverage and good visual representations?

To answer this question for conditional IGMs, we run experiments in the ImageNet100 setting and evaluate the linear classification performance for Gaussian views combined with pixel transformations, as we vary the number of unique samples. We show the results in Fig. 8. Note that we train all the encoders for the same number of iterations, equivalent to 200 epochs on ImageNet100 (matching the experiments shown in table 5 in App. E). This means, as the number of images (x-axis) increases, the number of times the model revisits a seen image decreases.

As Fig. 8 illustrates, the performance increases with more samples, but sub-logarithmically. These findings are consistent with recent work that found a small gap in generalization performance between online learning (infinite data) and a sufficiently large offline regime (Nakkiran et al., 2020).

## D FURTHER TRAINING DETAILS AND COMPARISON TO SIMCLRV1

We report in this section more training details for the unconditional baselines on real data, specifically SimCLR Chen et al. (2020c) training on ImageNet1000, to help clarify the reasons for the gap with Chen et al. (2020c).

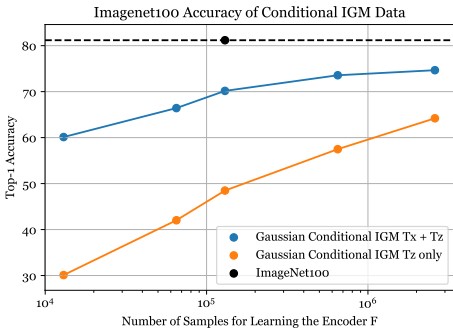

Figure 8: Effect of the number of samples for training the representation learner, evaluated using linear transfer to ImageNet100. "Gaussian" refers to the Gaussian views ($T_{\mathbf{z}} = \mathbf{z} + \mathbf{w}_{\texttt{Gauss}}$).

In our work, we train the image representations via contrastive learning using a batch size of 256 and evaluate them via linear evaluation using the same batch size. As such, the most comparable result in Chen et al. (2020c) would be $57.5\%/62.8\%$ from Table B.1, which is still higher than our reported $43.9\%$ in Table 1. We list below the main differences between our results and theirs.

1. **Lower image resolution**: Given that the IGMs we study in our work generate images, at $128 \times 128$, we train our representations on real data using the same resolution, both when training the encoder as well as when performing linear evaluation. This has a great effect on performance; in fact, when performing a linear evaluation using the same pre-trained encoder but with $256 \times 256$ images, the performance increases to $50.3\%$ for ImageNet1000, and $49.7\%$ for BigBiGAN with SimCLR pixel augmentations and gaussian latent augmentations.

2. **Optimizer in linear evaluation**: We use an SGD optimizer, as opposed to LARS (You et al., 2019), as proposed in Chen et al. (2020c). According to their experiments, linear performance drops from $62.8\%$ to $57.5\%$ based on the optimizer, with a lower effect when the batch size increases.

3. **Training time**: We train our encoder for 20 epochs, as opposed to 200 epochs in Chen et al. (2020c). To test the effect of training time, we train SimCLR on real data at 5, 10 and 15 epochs separately, to measure the effect of performance in transfer learning on ImageNet1000 classification with the same setup we described in Sec. 4 and we obtained $15.98\%$, $23.66\%$, $32.48\%$, respectively.

## E   FURTHER TRANSFER LEARNING TASKS

In this section, we further investigate the lighter weight protocol: train and evaluate representations only on data at the scale of ImageNet100 (Tian et al., 2019). This setting is described in the main text in Sec 4.

Moreover, we test on object detection task. For object detection, following (He et al., 2020), we use a Faster-RCNN (Ren et al., 2015) with the R50-C4 architecture. We fine-tune all layers for 24000 iterations on PASCAL VOC `trainval07+12`, with a batch size of 8. We report the standard COCO metrics, including $AP$ (averaged across multiple thresholds), $AP_{50}$, $AP_{75}$.

We report the results in Table 4 for unconditional IGM and Table 5 for the conditional IGM.

## F   MIXING REAL AND SYNTHETIC IGM DATA

Following (Ravuri & Vinyals, 2019), we test whether unsupervised models trained using real data can benefit from using data generated by an unconditional IGM. For these experiments, we use the ImageNet100 setting. Starting from ImageNet100, we replace a given percentage of the images by samples from BigBiGAN (Donahue & Simonyan, 2019), and a ResNet-50 using the SimCLR (Chen et al., 2020c) framework. For each model, we train a linear classifier on ImageNet100, using the

| Training Method | | | | Transfer Task | | | | |
|---|---|---|---|---|---|---|---|---|
| Data distribution | $T_\mathbf{z}$ | $T_\mathbf{x}$ | Objective | ImageNet100 Top-1 Accuracy | VOC07 Classification AP | VOC07 Detection AP | $AP_{50}$ | $AP_{75}$ |
| Real | – | SimCLR Augs. | Contrastive | 67.36 | 59.45 | 45.93 | 73.49 | 48.72 |
| Generated | – | SimCLR Augs. | Contrastive | 57.10 | 51.04 | 44.12 | 72.23 | 46.11 |
| Generated | $\mathbf{z}+\mathbf{w}_{Gauss}$ | SimCLR Augs. | Contrastive | **61.52** | **55.07** | **46.84** | **74.88** | **49.68** |
| Generated | $\mathbf{z}+\mathbf{w}_{steer}$ | SimCLR Augs. | Contrastive | 60.74 | 55.03 | 46.55 | 74.51 | 49.37 |
| Generated | – | – | Inverter | 28.36 | 22.95 | 39.56 | 66.38 | 40.56 |

Table 4: Results on **unconditional** IGMs. Real data is sampled from ImageNet1000 and distributed as $\mathbf{x} \sim T_\mathbf{x}(D)$. Generated data is sampled from BigBiGAN and distributed as $\mathbf{x} \sim T_\mathbf{x}(G(T_\mathbf{z}(\mathbf{z})))$. $T_\mathbf{z} = -$ indicates that no transformation is applied. For the *Contrastive* objective, positive and negative views are defined as described in Sec. 3.2.1 (without using class labels).

| Training Method | | | | Transfer Task | | | | |
|---|---|---|---|---|---|---|---|---|
| Data distribution | $T_\mathbf{z}$ | $T_\mathbf{x}$ | Objective | ImageNet100 Top-1 Accuracy | VOC07 Classification AP | VOC07 Detection AP | $AP_{50}$ | $AP_{75}$ |
| Real | – | SimCLR Augs. | Sup. Contrastive | 81.18 | 65.79 | 45.76 | 74.72 | 48.58 |
| Generated | – | SimCLR Augs. | Sup. Contrastive | 66.82 | 56.41 | 43.87 | 72.88 | 45.50 |
| Generated | $\mathbf{z}+\mathbf{w}_{Gauss}$ | SimCLR Augs. | Sup. Contrastive | **70.16** | **58.26** | 44.23 | 73.51 | 45.73 |
| Generated | $\mathbf{z}+\mathbf{w}_{steer}$ | SimCLR Augs. | Sup. Contrastive | 67.86 | 57.47 | **44.49** | **73.31** | **45.85** |
| Generated | $p_\mathbf{z}$ | SimCLR Augs. | Sup. Contrastive | 68.08 | 57.94 | 44.08 | 73.06 | 45.49 |
| Generated | – | – | Inverter | 38.84 | 24.93 | 42.02 | 69.29 | 43.62 |

Table 5: Results on **class-conditional** IGMs. Real data is sampled from ImageNet1000 and distributed as $\mathbf{x} \sim T_\mathbf{x}(D)$. Generated data is sampled from BigGAN and distributed as $\mathbf{x} \sim T_\mathbf{x}(G(T_\mathbf{z}(\mathbf{z}), y))$. $T_\mathbf{z}, T_\mathbf{x} = -$ indicates that no transformation is applied. $T_\mathbf{z} = p_\mathbf{z}$ indicates that the transformation draws a new sample $p_\mathbf{z}$, independent of the original $\mathbf{z}$. For the *Sup. Contrastive* objective, positives are defined following Sec. C.1.1, where two views are treated as positive if and only if they share the same label $y$.

| Training Method | | | | Transfer Task | | | | |
|---|---|---|---|---|---|---|---|---|
| Data distribution | $T_\mathbf{z}$ | $T_\mathbf{x}$ | Objective | ImageNet100 Top-1 Accuracy | VOC07 Classification AP | VOC07 Detection AP | $AP_{50}$ | $AP_{75}$ |
| Real | – | SimCLR Augs. | Classifier | 80.64 | 67.99 | 45.90 | 75.08 | 48.24 |
| Generated | – | SimCLR Augs. | Classifier | 65.18 | 62.41 | 43.45 | 72.88 | 45.46 |

Table 6: Results on **class-conditional** IGMs. Real data is sampled from ImageNet1000 and distributed as $\mathbf{x} \sim T_\mathbf{x}(D)$. Generated data is sampled from BigGAN and distributed as $\mathbf{x} \sim T_\mathbf{x}(G(\mathbf{z}, y))$. We report the performance on real data and synthetic data with no latent transforms, for the Supervised Contrastive and Classifier objectives.

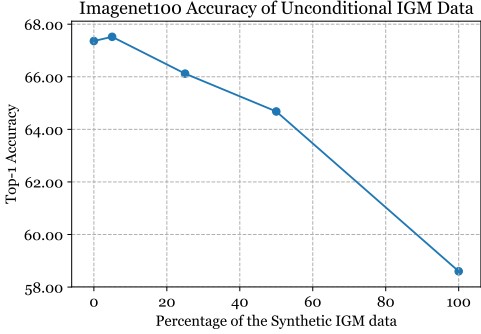

Figure 9: Mixing real data and synthetic IGM data as a way of augmenting real data. The plot shows the percentage of real data replaced the fake one versus the top-1 percent accuracy.

learned features, and report Top-1 accuracy. Similar to (Ravuri & Vinyals, 2019), we see a decreasing trend in the performance as the number of real images decreases, but find a sweet spot in using a small percentage (5%) of synthetic images.

# G  VISUALIZATION OF LATENT TRANSFORMATIONS

Here we present qualitative examples of our methods in latent view creation for both unconditional and class-conditional IGMs as illustrated in Fig. 10 and Fig. 11, respectively.

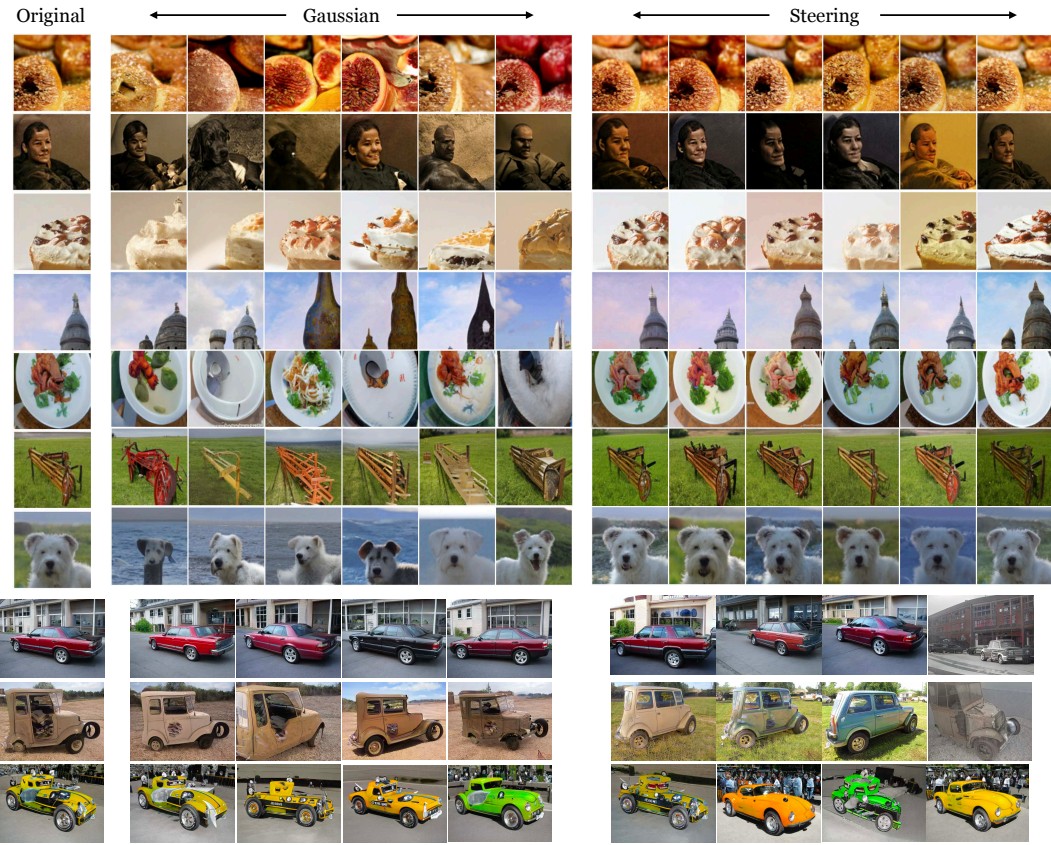

Original     ←——— Gaussian ———→     ←——— Steering ———→

Figure 10: Examples of different latent transformation methods for unconditional IGMs data using BigBiGAN and StyleGAN LSUN CAR (the last three rows). Columns from left: anchor, Gaussian neighbors, and steering neighbors.

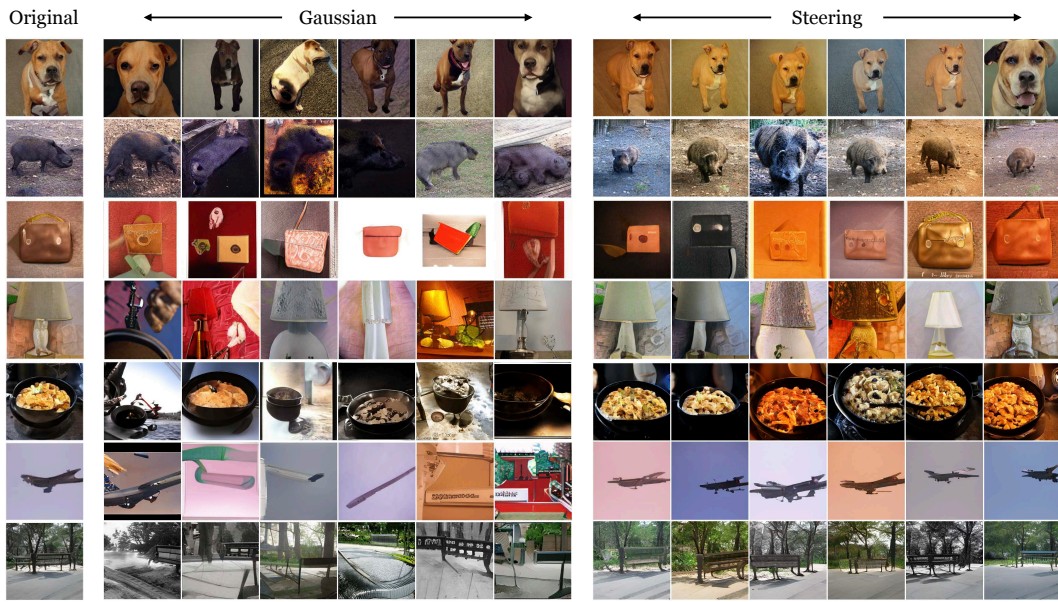

Original     ←——— Gaussian ———→     ←——— Steering ———→

Figure 11: Examples of different latent transformation methods for conditional IGMs data using BigGAN. Columns from left: anchor, Gaussian neighbors, and steering neighbors.

