# OpenReview forum: "Generative Models as a Data Source for Multiview Representation Learning"
_ICLR.cc/2022/Conference — ICLR 2022 Poster_

### Official Review · Reviewer_P9Ys · 2021-10-21

**Correctness:** 3
**Technical Novelty And Significance:** 3
**Empirical Novelty And Significance:** 3
**Recommendation:** 6
**Confidence:** 4

**Main Review:**

**Strengths**

* The paper is well motivated. The authors tackle an important and relevant problem, how to use samples from a generative model in a meaningful way for downstream tasks.
* To the best of my knowledge, this is the first paper that attempts to utilize black-box generative models for unsupervised representation learning.
* The entire paper is easy to read and self-contained. The literature review is also very solid.

**Weaknesses**

The major weaknesses of the papers are the empirical results. Here are my concerns:

* It seems that the SimCLR baseline is very weak. The lowest reported number in Chen et al 2020 is 62.8% ImageNet top 1 accuracy using a batch-size of 256 and a number of epochs equal to 100 while the paper reports 43.9%. What are the differences in this paper as compared to the SimCLR baseline? Even though achieving state-of-the-art performance is not the goal of this paper, the authors should strive for a reasonable baseline.
* The authors could try to provide more insight on their negative result which is that the contrastive algorithm trained on generated views underperforms the SimCLR baseline by 1.3%. Is it that the generated samples have poor quality as compared to the ground truth dataset, or do they lack in diversity (for eg, the typical mode collapse issue with GANS)?
* **Steered Latent Views**: It seems that $w_{steer}$ in $z + w_{steer}$ is optimized to match pixel-level transformations targets $T_z(z)$. Then why not use the pixel-level transformations directly? Can the authors clarify the setting where it is preferred to use $T_z(z + \alpha w)$ instead of $T(G(z), \alpha)$?
* The baseline for the real algorithm in Table 2 is missing.
* The ImageNet100 benchmark is not apples-to-apples since both the “real” and “generated” algorithms have access to samples with a varying number of classes. Could the authors clarify the motivation behind using ImageNet100?
* The authors should also plot the "real baseline" bound in Figure 6, to see if the real baseline can be approached simply by increasing the number of samples.

Typos:
* Legends in Figure 6 have the same color.
* Eq 6) should also have T_z?


**Summary Of The Paper:**

The paper explores applying a black-box generative model (instead of a fixed training dataset) to learn unsupervised visual representations, mainly using contrastive learning. Standard contrastive learning algorithms generate multiple views of a datapoint using transformations in pixel space. The authors explore using transformations in latent space, either in isolation or in combination with pixel-level transformation methods.

Results are reported for two generative models.
* BigBiGan trained on ImageNet and finetuned on ImageNet100 and ImageNet1000
* StyleGan2 trained on LSUN Car and finetuned on ImageNet


**Summary Of The Review:**

The direction that the authors pursue in this paper is challenging and novel. However, the results are mainly negative.

For negative results, to be useful to the community, imho either the results should be suprising or should offer some insight. I have given a reject rating mainly because of the empirical results.

Please see the weaknesses section above. If atleast the first two bullets points can be addressed convinvingly by the authors, I can reconsider my rating.

**Update after Rebuttal**:

I thank the authors for their additional experiments and discussion. I've upgraded my rating from 3 (Reject) to 6 (Lean Accept).
The rationale for my rating change:

* The authors confirmed that the SimCLR baseline is lower than the reported results, due to the low resolution (128x128) of the generated model and reduced training time.
* The models (baseline and generative) are trained for 20 epochs using cyclic learning rates so I think the results might be transferrable to full training.
* The authors have shown more promsing results on pretrain LSUN + transfer on Stanford Car classification.
* ImageNet100 results as far as I see are provided only in Section 4.1.2 and 4.3 as a diagnostic tool, so the comparisons are fair.

The authors have taken a first-step towards usage of generative models for unsupervised representation learning. Even though, the results are somewhat underwhelming on ImageNet itself, this work can serve as a promising and strong baseline for future work in this topic.

**For the final version**
If the paper gets accepted, I encourage the authors:
* To report results for full training (100 Epochs), at least with the "Real baseline"  and their best setting for generated data
* To open-source their code.
* To explicitly state that the latent transform is applied only once, as compated to pixel-level transformations, which are applied twice. This might be a trivial detail but can be extremely important to reproduce the authors results.

---

> ### Author Response · Authors · 2021-11-16
> **Author's response**
>
> Thank you for your detailed comments and helpful suggestions. We address your specific questions and concerns below.
>
> **Low SimCLR Baseline:**
> This is a good point, and we will make the distinction more clear in the manuscript. While the batch size in our setting and in the mentioned result from Chen et al. 2020 is the same (256), there are two main differences between our settings: 1) we train the encoder for 20 epochs, as opposed to 100 epochs, and 2) we use images of half the resolution (128x128) to train both the contrastive representations as well as the linear classifier, which explains the drop in performance.
>
> Given the time constraints in the rebuttal, we trained the linear classifier on 256x256 images (keeping the encoder trained at 128x128), to partially measure the effect of resolution. The ImageNet accuracy improves to 50.3%, even though the encoder is still trained at 128x128. Note that training the encoder at 256x256 would be an unfair comparison to the unsupervised IGM results, since the pre-trained BigBiGAN can only generate images at 128x128.
>
> We could have trained the linear classifiers at the larger resolution, which would improve the performance, but we would argue that this is not a technically sound approach, given what we are studying in this paper: given that we are using the encoder to extract features that we linearly evaluate, it makes sense that the resolution of the input images matches what the encoder was trained with.
>
>
> **More insights on negative results:**
> We would like to point, as described in the general response, that the fact that these generative models underperform representations learned with real data is somewhat expected, and not necessarily a negative result. While we provide comparisons with real data as a reference, the main purpose of this paper is to explore the settings where it is not even possible to access the real data a generative model is trained on. Our work provides insights under this setting, including how we can leverage walks in latent space for data augmentation, how these models transfer to new tasks and datasets and what are the effects of using larger amounts of generated data.
>
> The above being said, studying the reasons for the gap with real data is an interesting question, as it might inform how to build better generative models. Our paper shows that the performance of representations learned with IGMs increases sub-logarithmically with the number of synthetic training data, hinting at the fact that even if we increase the amount of synthetic data, its diversity does not increase.
>
>
> **Using pixel transformations vs steered latent views:**
> While $w_{steer}$ is optimized to match pixel-level transformations, walking in the latent space along these directions also changes the semantics of the image so that the image generated from the new latent code is still plausible. You can refer, for example, to Figure 4, second column of [1], where a brighter version of the volcano image leads to changing the state of the volcano, and time of the day. It would not be possible to generate these changes with pixel level transforms.
>
> [1] Jahanian, Ali, Lucy Chai, and Phillip Isola. "On the" steerability" of generative adversarial networks." International Conference on Learning Representations. 2019.
>
>
> **On Imagenet100:**
> We agree that the comparisons in Imagenet100 between the real and synthetic cases are not totally fair. We used Imagenet100 as a way to speed up experimentation on different parameters for the transformations or study the effect of dataset size, but our main results are on Imagenet1K. For Imagenet100, we provide the results on real data as a reference, but, particularly in the unconditional case, they are not totally comparable since the IGM model could be generating images from classes out of the 100-class set. We want to emphasize though that both encoders are trained with the same number of images, even though the IGM dataset may be more diverse for the aforementioned reason.
>
> **Correcting Figure 6:**
> Thank you for the suggestions, we have updated the plot with the real baseline, and updated the color of the legends. Note that the ImageNet100 performance of the unconditional IGMs does get really close to the ImageNet100 performance with enough samples, but as stated above, it is not a totally fair comparison since the unconditional IGM model was trained on ImageNet1K. In the conditional case (Fig 8.) the model only sees IGM images from the 100 classes, and the gap to real is more clear.
>
> **Equation 6:**
> In our setting, we only apply latent transforms on the positive and negative samples, keeping the $z_{a}$ from the anchor sample fixed. We could have transformed the anchor as well, though, as the reviewer suggests.

---

> > ### Comment · Reviewer_P9Ys · 2021-11-19
> > **Followup Questions**
> >
> > Important: Note that I do not see any updates yet in the current PDF. I strongly urge the authors to update the manuscript with the  differences in their experimental setup as compared to Chen et 2020 before the rebuttal deadline (as mentioned in the response)
> >
> > Thanks for the response! I have a few follow-up questions:
> >
> > Impact of training time
> > ----------------------------------
> > Thanks for clarifying that the results (in part) are worse than Chen et al. 2020 due to the lower image resolutions 128x128 generated by BigBiGAN. Can the authors provide more insight on the impact of training time? How long does it take to run 100 epochs and why was the model undertrained? If the model cannot be trained before the rebuttal deadline, I suggest the authors could also show the progress of downstream accuracy as a function of training time, to see if both variants (static dataset and dataset from generative model) benefit equally from training longer.
> >
> > Number of views
> > -----------------------------
> > > In our setting, we only apply latent transforms on the positive and negative samples, keeping the  from the anchor sample fixed. We could have transformed the anchor as well, though, as the reviewer suggests.
> >
> > Can the authors comment more on why this decision was made? In 3.2.1, the anchor was transformed as well while doing pixel-wise transformations. Why are the anchors not transformed while doing latent-space transformations. Did the authors try it and observe worse results?
> >
> > Can the authors also clarify that the SimCLR algorithm used s exactly the same as in Algorithm 1 of Chen et al 2020? In particular, if the loss term is sum of 2*K sub-loss terms (for both the baseline and their model trained on samples), where K is the minibatch size?
> >
> > On ImageNet100:
> > -----------------------
> > My comments about this are not that important as compared to the other comments. If ImageNet100 is mainly used as a test-bed, I would restructure the experiments section such that 1) First the experiments on ImageNet100 are introduced as a synthetic test-bed 2) Then the experiments on the "real ImageNet dataset"
> >
> > **Nitpick**: Is the ImageNet1000 just ImageNet? I would suggest to replace all ImageNet1000 occurences with ImageNet in the case.
> >
> > Thanks for the clarifaction on the steered views. It is indeed fascinating that while $w_{steer}$ is optimized to match pixel-level transformations, the walks in this direction are more realistic.

---

> > > ### Author Response · Authors · 2021-11-23
> > > **Author's response: Thank you for your followup questions and comments. We address your specific questions and concerns below.**
> > >
> > > Thank you for your comments! We have updated the manuscript with a section in the Appendix C, pointing out the main differences between our setting on real data and the one on SimCLR, that we think may explain the gap. We refer to the appendix on page 7.
> > >
> > > Although our experimental setting does not quite match SimCLR, we use a consistent setting between all experiments which allows for direct comparison of what works and what doesn't amongst the methods we evaluate. We find that latent space augmentations, T_z, consistently improve performance across the settings we tested.
> > >
> > > *Impact of training time*
> > > Our SimCLR model trained on 128x128 images took 18 hours to train up to 20 epochs, requiring around 8 days to train for the full 200 epochs, with 24h to train the linear classifier on top. While this is a reasonable time for a single experiment, this setting would have required us to train for that amount our models on different sources of data and transformations. While we are not on time to train for 200 epochs, we are currently training our models (both IGM and real) at 5, 10 and 15 epochs, to measure the effect of performance and we aim to report them as we obtain them. So far, for real data, we have obtained 23.66% and 31.63% for 10 and 15 epochs, respectively. Note that we cannot reuse the intermediate weights of our previously trained model, since the learning rate follows a cosine schedule that depends on the number of epochs, making the intermediate checkpoints not representative.
> > >
> > > *Impact on number of views*
> > > We did not try applying the latent transforms in both images, though it would be worth exploring this. Note that we would likely have to choose a different std for the transform, since images might be farther in the latent space than in our setting.
> > >
> > > As the reviewer notes, we used the same setting as in Chen et al, computing the loss as the sum of 2K loss-terms, with K being the batch size.
> > >
> > > *On Imagenet100*
> > > Thank you for the suggestion. While ImageNet100 is a great test-bed for experimentation (including measuring performances with dataset size), it has the previously described limitation in that the generative models are still trained with ImageNet1000, and as such the comparison with real data is not totally comparable. Our paper aims to study the representation capabilities of IGMs trained on the full ImageNet dataset, and this is why we thought it was important to start with those results, followed by a more detailed study of our space of transformations and dataset size on a synthetic (IGM) dataset.

---

> > > > ### Comment · Reviewer_P9Ys · 2021-11-23
> > > > **Rating Update**
> > > >
> > > > Thanks! I've updated my rating.

---

> ### Author Response · Authors · 2021-11-23
> **Author response to "the results are mainly negative"+"The baseline for the real algorithm in Table 2 is missing."**
>
> We ran this experiment and found that real achieves 40.70%, while StyleGAN2 IGM achieves 49.79% (as was previously reported in Table 2 for Stanford Car Classification).
>
> In this setting, our results are clearly positive: IGM data substantially outperforms an equal number of real data samples. This is what we expect as IGMs become better (and discussed in the paper).
>
> Details: To run this experiment, we needed the 893K subset of real images from LSUN Car that trained StyleGAN2. The LSUN Car dataset contains 5+e6 images. However, because we could not find that subset, we used StyleGAN2 dataset_tool.py to reproduce a subset of LSUN Car. This way we collected 893K images. We used this dataset as real data to train SimCLR. We then tested the encoder on the Stanford Car model classification.
>
> We will continue this experiment instead by training StyleGAN on our subset to make sure we use the exact images for both real and IGM setups. We will continue updating our results on this website as we obtain the results.

---

### Official Review · Reviewer_ZFjV · 2021-11-01

**Correctness:** 4
**Technical Novelty And Significance:** 3
**Empirical Novelty And Significance:** 4
**Recommendation:** 8
**Confidence:** 3

**Main Review:**

Strengths:
- The problem of learning with only generated data is very relevant, and the authors achieve reasonably good results.
- The methods are very clearly exposed, the results are comprehensive and the experimental protocol is very detailed.
- It should be appreciated that while the paper is about benchmarking, it makes an effort to play to generative models' strength by also using latent space augmentations for contrastive learning. This approach is novel and the empirical results support its importance.

Weaknesses:
- It might be a little bit misleading that much of the paper is about implicit generative models but only GANs are tested. Especially since while the general idea is universally applicable, the specifics, and in particular latent augmentation, are not necessarily trivial to transferred to other generative models.


**Summary Of The Paper:**

The paper investigates if synthetic datasets obtained from implicit generative models can be used for representation learning in place of the original dataset.
To do so, they compare the performance of multiple popular contrastive learning methods trained on real data against the same methods trained on synthetic data generated by GANs.
Notably, for the generated data, in addition to standard pixel transformation, they also evaluate latent space augmentations methods for contrastive learning.

**Summary Of The Review:**

The paper is a step forward towards training only using synthetic data, a long-standing goal that has been of interest since the introduction of deep generative models.
Moreover, it introduces and evaluates a novel way of combining generative data and contrastive learning that is shown to be very promising.
With the rapid advances in representation learning, this submission providing a strong and comprehensive empirical study is likely to be valuable to the community.

---

> ### Author Response · Authors · 2021-11-16
> **Author's response**
>
> Thank you for your comments and suggestions. We are encouraged that you find the problem relevant and value the clarity of our experiments and the novelty of our approach. We address your specific questions and concerns below.
>
> **Study limited to GANs**
> GANs are one of the most popular and effective IGMs, with multiple pre-trained models  and  a  disentangled  and  smooth  latent  space,  and with well-studied latent transforms mapping nearby points to different views of an image. While we focused on GANs to keep the scope of our work tractable, notice that many of our ideas could be applied in other IGMs with small changes. We agree with the reviewer that studying these representations in other families of IGMs merits further study. We leave those for future work.

---

### Official Review · Reviewer_ZWWA · 2021-11-01

**Correctness:** 4
**Technical Novelty And Significance:** 2
**Empirical Novelty And Significance:** 3
**Recommendation:** 8
**Confidence:** 4

**Main Review:**

### Strengths
- The paper is well-written and easy to follow. The proposed method is described and illustrated clearly and appears technically sound.
- Overall, the experiments and ablations systematically support the effectiveness of the proposed method. Both, unconditional and class-conditional representation learning is considered.
- Using generative models as a data source is an interesting idea that is gaining attention recently as it might be beneficial to compress excessively large datasets, address privacy concerns, and enable more diverse data augmentation for training.

### Weaknesses
- A highly similar idea to the proposed latent-space transformations for contrastive learning was already proposed by Yang et al. [1] as “Noise perturbation”. This work is not referred to in the paper. While this work focuses on more stable GAN training and not representation learning, it takes away from the novelty of the proposed method.

> [1] Ceyuan Yang, Yujun Shen, Yinghao Xu, Bolei Zhou. Data-Efficient Instance Generation from Instance Discrimination. arXiv preprint arXiv:2106.04566, 2021.

### Additional questions / comments:
- As the performance of the proposed method probably largely depends on the quality of the generated images, an ablation considering the truncation value would be interesting. I.e., what is more important for representation learning: image diversity or image fidelity? Also, please report how the truncation parameter was determined for the existing experiments.
- What are the details for evaluating the BigBiGAN encoder? In the paper, you mention that it is trained on images of size 256x256 while you use resolution 128x128. Does this mean you resize the images before showing them to the BigBiGAN encoder? Further, please add the result of this experiment to the corresponding table (Table 4) rather than in the text.
- For the ablation in Figure 6: Why does it make sense to keep the number of iterations constant here? Would you not be rather interested in the best possible performance, i.e. training each model for the best validation error with the same number of sufficient epochs for convergence?
- What is your recommendation for tuning the standard deviation? Did you perform a grid search for each dataset, similar to the ablation on ImageNet1000 in Figure 5?
- Why does steering perform better than Gaussian perturbation in the class-conditional setting (Table 3)?
- ContraD (https://arxiv.org/pdf/2103.09742.pdf) combines contrastive learning with GAN training and finds that training a joint discriminator can be beneficial for both tasks, image synthesis, and representation learning. This should be added and put into context in your related work.
- Another interesting related work where generative models are used as a data source, to generate multiview images and allow for efficient data labeling: https://nv-tlabs.github.io/GANverse3D/. Should be added to related work.

### Misc:
- Table 3: bold number for ImageNet1000 missing

**Summary Of The Paper:**

The work investigates **generative models as a data source for self-supervised representation learning**. In particular, the authors propose to form contrastive pairs in the latent space of the generative model and combine it with standard contrastive learning where pairs are formed in image space.
While performance is **slightly inferior** to training with the full dataset, the proposed approach **only requires storing the model weights** of the generator instead of the full dataset.

**Summary Of The Review:**

The considered task, i.e. generative models as a data source for representation learning, is **very interesting for the community** and has recently received increasing attention. The experiments in the paper appear sound and provide **interesting insights**. My major concern is about **the contribution regarding the latent space transformations**, as a very similar strategy was already proposed by Yang et al. Nonetheless, the findings of this work are interesting, so currently I am leaning towards accepting the paper.

**Post-rebuttal**:
The authors successfully addressed my main concern about the paper by Yang et al. so I update my rating from 6 to 8.

---

> ### Author Response · Authors · 2021-11-16
> **Author's response**
>
> Thank you for your detailed comments and helpful suggestions. We address your specific questions and concerns below.
>
>
> **Related work:**
>
> [Edit]: We have revised the manuscript with the suggested references. Thank you again for the pointers.
>
> Thank you for the pointers to related works, we will cite them and put them in context in the revised manuscript. Yang et al. [1] explore the effect of latent space transformations to improve image generation, which is consistent and complementary with our results, where we show better learned representations. Note that this work is concurrent with ours, as well as complementary with our studies. Moreover, we study these effects in a larger set of transformations and conditions. ContraD incorporates fake samples in a supervised contrastive objective, but is different to our setting in two key aspects: 1) it assumes access to a dataset of real images, as opposed to one of the main premises in our paper. 2) it only creates different views for real images, without exploring how to create views from IGM images. GANVerse is also a relevant related work, suggesting interesting future work in disentangling the latent space for better augmentations. We explore that direction to a certain extent with our steering transforms.
>
>
> **Changing truncation values:**
> This is a good suggestion. Indeed, there is a sweet spot in the truncation parameter that allows for a balance between diversity and quality for representation learning. We note that this effect is already studied to a certain degree in Ravuri et al. 2019, showing that larger truncation values decrease the inception score but improve classifier accuracy. Following their work, we chose a large truncation value to ensure enough diversity.
>
>
> **Details of BigBiGAN encoder evaluation:**
> For the BigBiGAN experiments, we train a linear classifier over features extracted from the BigBiGAN ResNet-50 encoder. The input images are 256x256, which gives a disadvantage to our proposed models. Still, the linear classifier performance is lower than in the supervised contrastive methods we explore in the paper. We are not including this result in Table 4, since the conditions are slightly different.
>
>
>
> **Ablations in figure 6:**
> Notice that when changing the amount of training samples, we cannot really make comparisons in terms of number of epochs, since more training samples will mean having more images per epoch and therefore the model will be training for longer than when having less training samples. Here we are interested in studying the effect of dataset size, rather than amount of training, which is why we keep the number of iterations (gradient updates) the same.
>
>
> **Tuning the standard deviation:**
> Similar to the truncation parameter, selecting an optimal standard deviation for each dataset would require a grid search with a validation set to measure the effect in performance. In Figure 5, we note the effect of this standard deviation on ImageNet100, drawing parallelisms with InfoMin results from Tian et al. 2020.
>
>
> **Steering performs better than Gaussian:**
> Note that the performance with steering transforms is actually really close to the one with Gaussian transforms, and it may be that their performance is comparable. The fact that steering performs comparatively better than in the unconditional case may be because the transforms change the aspect of the image while keeping the class the same, while in the unconditional case it may be that the steering transforms the contents of the image too much.
>
>
> **Bold number missing:**
> Thank you for the note, we have updated the table in the revised manuscript.

---

> > ### Comment · Reviewer_ZWWA · 2021-11-30
> > **Response to Rebuttal**
> >
> > Thank you for putting the related works into context and for explaining the details I was asking for. This resolves my main concern and I updated my rating accordingly.

---

### Official Review · Reviewer_pxGz · 2021-11-08

**Correctness:** 4
**Technical Novelty And Significance:** 2
**Empirical Novelty And Significance:** 4
**Recommendation:** 8
**Confidence:** 4

**Main Review:**

This is an exploratory study of the effects of performing representation learning based on a generative model trained on data, as opposed to the data itself. It is very well written and coherently laid out, and was a pleasure to read. There are no theoretical advances, but the paper provides a useful first foray into an interesting topic, based on well executed empirical studies.

Strengths: the paper provides a well reasoned empirical intro into a fairly novel area. The experiments appear to be well executed and thought through. The results are not earth shattering, but solid work like this should be published and discussed. The concept of using latent space perturbations rather than pixel level perturbations is an interesting (in a good way!) one. One area of further research might be into how one chooses these perturbations in a disciplined way, and how these choices influence the mapping from one latent space to another (the learned latent space). In extremis, would this result in latent space 'cloning'?

Weaknesses: The elephant in the room is that the results depend on the quality of the IGM (implicit generative model) itself - by quality i mean how well has the IGM learnt the data-generating distribution. This of course is a function of both the model's expressivity as well as the complexity of the data-generating distribution (and how well / evenly the training data covers the distribution). The paper makes the claim that the IGM can be considered a compact representation of the data itself - were this true, the performance of models trained on the IGM would be near to indistinguishable from the performance of models trained directly on the data, which isn't the case.
I was less surprised than the authors that Gaussian perturbations in the latent space performed as well as more reasoned perturbations using steering methods - I think because I assume that how we think about data may not be as useful as we think (!); indeed much of modern machine learning bears this out.

**Summary Of The Paper:**

This is an exploratory study of the effects of performing representation learning based on a generative model trained on data, as opposed to the data itself. It is very well written and coherently laid out, and was a pleasure to read. There are no theoretical advances, but the paper provides a useful first foray into an interesting topic, based on well executed empirical studies.

**Summary Of The Review:**

No theoretical advances, but some clever thinking and solid results. Happy to recommend acceptance.

---

> ### Author Response · Authors · 2021-11-16
> **Author response**
>
>
> Thank you for your comments, we are encouraged that you find our empirical study valuable and well executed, and well as our exploration of latent space augmentations for representation learning. Indeed, the quality of the results depend on how well the IGM has learned the data distribution and its capacity to represent diverse data. As noted by the reviewer, the studied IGMs cannot fully represent the original dataset, which explains why the IGM representations are slightly below the real ones. As more IGMs are developed, we hope their learned representations can compete with those learned from real data.
>
> **Exploring more latent transformations**
> This is indeed an exciting area for future research. One possibility could be to learn the space of latent transformations from the contrastive loss, allowing to create hard negative samples given an anchor image to learn stronger representations.
>
> **Gaussian vs Steering perturbations**
> Indeed, the fact that steering methods produce more visually interpretable transformations does not necessarily mean that they are better suited to learn strong representations, as we observe in our results. Learning transforms that lead to representations with better performance on downstream tasks is an interesting direction of future research.

---

### Comment · Area_Chair_ws9B · 2021-11-16
**Paragraph about ethics and social impact**

Please consider adding a statement about ethics and social impact of this paper given the use of generative models as a data source. Thank you!

--AC

---

> ### Author Response · Authors · 2021-11-16
> **Paragraph about ethics and social impact**
>
> Thank you for the note, we will update the paper with an ethics and social impact paragraph shortly.
>
> [Edit]: We have uploaded a revised version, including an ethics statement

---

### Author Response · Authors · 2021-11-16
**To everyone**

To all:

We thank reviewers for their insightful feedback. We are encouraged that reviewers consider the proposed problem relevant to study, and that they find the paper clear and the empirical studies well motivated. We have updated the paper with the reviewer’s comments and suggestions, and added an ethics statement section.

We want to emphasize that the purpose of this paper is to study what are the best approaches and properties of learned representations when there is no access to real data. We do not expect these representations to surpass the representations learned from real data yet. As pointed out by the reviewers, IGM generated images are lower in quality than real images, can collapse into certain modes, or fail to be able to represent certain visual concepts [1]. However, given the success and trend in developing models trained with massive amounts of data, the scenario where these models are provided as black-box interfaces, as opposed to providing the training data, is becoming increasingly common (GPT-N, DALL-E, CLIP). For this reason, it is important to study the properties of models under this setting, where access to real data is not a possibility. Our paper provides insights into what properties we should exploit under these settings, and what kind of behaviors we could expect from these models.

[1] Bau, David, et al. "Seeing what a gan cannot generate." Proceedings of the IEEE/CVF International Conference on Computer Vision. 2019.

---

### Decision · Program_Chairs · 2022-01-20

**Decision:**

Accept (Poster)

**Comment:**

All of the reviewers appreciate the clarity of exposition and the importance of the problem studied. That said, I agree with Reviewer P9Ys that the results are somewhat underwhelming. The baselines appear weak and are likely not well tuned on the Stanford car dataset. Key question that remains unanswered in my opinion is whether this is the most effective approach to using synthetic data to improve classification accuracy (e.g., in contrast to [Ravuri & Vinyals, 2019](https://arxiv.org/abs/1905.10887) and follow-up work). Nevertheless, I believe the community will benefit from this paper's contributions and this line of work.